# Benchmarking Efficiency Techniques in GenAI Foundation Models Using an Elo-Based Performance Evaluation Framework

Saman Keon
Meta AI

Summer Deng
Meta AI

Bram Wasti
Meta AI

Josh Fromm
Meta AI

*Abstract*—**Evaluating the effectiveness of efficiency techniques in foundation models—such as quantization, pruning, and distillation—requires a rigorous, standardized methodology for determining model quality parity. Notably, quantization poses a unique challenge due to its non-obvious impacts on model quality stemming from alterations to numerical representations, which are not captured by established scaling laws. In this work, we address this critical gap by proposing an Elo-based scoring framework that quantifies the relative performance of optimized models through automated competitive matchups. By leveraging publicly available datasets such as LMSYS chat, which encompass diverse language-based real-world user queries, our method generates consistent and interpretable rankings of model variants using LLM-based preference judgments. This approach enables quality assessments across various tasks without relying on task-specific ground truths. Backed by over 2,000 GPU hours on H100 infrastructure, our framework offers a scalable, reproducible evaluation protocol that delivers nuanced insights into the trade-offs of model efficiency techniques, while taking a step toward standardizing performance parity assessment across the machine learning community.**

## I. INTRODUCTION

The growing demand for deploying foundation models in real-world applications has amplified the importance of efficiency techniques that reduce computational costs without sacrificing model quality. Methods such as quantization, pruning, and knowledge distillation have proven essential for enabling scalable and sustainable AI systems, particularly for resource-constrained environments. However, evaluating the impact of these techniques remains challenging. Traditional task-specific metrics, while valuable, often fail to capture subtle quality degradations that efficiency optimizations can introduce, especially in generative or open-ended settings.

Quantization, for instance, alters the underlying numerical representations of model parameters, leading to non-trivial and sometimes unpredictable effects on output quality. These effects are not well-characterized by established scaling laws or standard benchmarks, creating a critical gap in our ability to rigorously assess optimization trade-offs. Moreover, the lack of task-agnostic, scalable, and interpretable evaluation frameworks further complicates fair comparisons between model variants optimized for efficiency.

Recent advances in preference-based evaluation, particularly using Elo-style ratings derived from large-scale pairwise comparisons, offer a promising alternative. Frameworks like LM-SYS Chatbot Arena have demonstrated the viability of using LLM-as-a-judge approaches to evaluate model quality based on human-aligned preferences rather than rigid task-specific correctness. Building on this insight, we propose an Elo-based performance evaluation framework tailored to systematically benchmark the effects of efficiency techniques—starting with quantization—on foundation models.

Our contributions are threefold: (1) we introduce a scalable, model-agnostic evaluation protocol that leverages LLM-based judgments to generate interpretable Elo scores across model variants; (2) we validate our approach through extensive experiments on quantized versions of the `google/gemma-3-1b-it` model, using over 2,000 GPU hours of evaluation on real-world prompts; and (3) we provide insights into the stability, transitivity, and computational efficiency of preference-based evaluations, laying the groundwork for future benchmarking efforts across the broader landscape of model efficiency techniques.

## II. BACKGROUND AND MOTIVATION

Recent advances in foundation models have pushed the boundaries of performance across a wide range of tasks in natural language processing (NLP), computer vision (CV), and multimodal reasoning. However, these gains have come at a significant computational cost, making efficiency techniques such as quantization [6] [15], pruning [3] [8], and knowledge distillation [4] [11] increasingly essential for enabling real-world deployment. While these methods are effective at reducing latency, memory footprint, and energy consumption, evaluating their impact on model quality remains an open challenge. For instance, quantization introduces changes to numerical representations that can have non-linear and unpredictable effects on model behavior [1] [14]. These effects are not well-captured by traditional evaluation metrics like perplexity, BLEU, or accuracy, which typically rely on task-specific ground truths and may obscure subtle degradations in quality, particularly in generative or open-ended tasks [10]. Moreover, existing model evaluation methodologies often fail to generalize across tasks or modalities. Standard benchmarks such as GLUE [12], SQuAD [9], and ImageNet [2] are useful for targeted evaluations, but they provide little insight into how models behave in unconstrained, user-facing settings. Scaling laws [7] [5], while powerful for understanding model behav-

ior during training, offer limited guidance post-optimization, especially for quantized models that diverge from floating-point training dynamics. Human preference judgments have emerged as a valuable tool for evaluating generative models, particularly in the context of large language models (LLMs). Recent work, including the LMSYS Chatbot Arena [17], has demonstrated the viability of using pairwise comparisons to generate Elo-style rankings of models based on their perceived quality across diverse user queries. Elo scoring offers several advantages: it is interpretable, incrementally updatable, and grounded in relative performance rather than absolute correctness. This makes it well-suited for scenarios where task-specific ground truths are unavailable or insufficient. Despite these promising developments, there is currently no standardized framework for using preference-based evaluation to benchmark the effects of efficiency techniques across foundation models. Our work addresses this gap by proposing a multimodal, Elo-based evaluation framework that leverages LLM-based preference judgments on publicly available datasets. By doing so, we aim to provide a consistent and scalable method for understanding the quality trade-offs introduced by efficiency techniques—especially in cases where traditional metrics fall short.

## III. METHODOLOGY

In this section, we describe the design of our Elo-based evaluation framework, which enables standardized quality comparisons of efficiency-optimized language models through preference-based matchups. Our approach is model-agnostic, scalable, and designed to reflect real-world user utility in the absence of task-specific ground truth.

### A. Overview

The core idea behind our methodology is to assess the *relative* quality of models using **pairwise comparisons** of their outputs in response to a shared prompt. These outputs are then judged by a third-party large language model (LLM-as-a-judge), and the results are used to update Elo scores, producing a ranked leaderboard of model variants. This structure allows for interpretable and statistically robust comparisons across quantized models without relying on task-specific accuracy metrics.

### B. Model Variants and Quantization Techniques

We evaluate a set of quantized variants of the google/gemma-3-1b-it model using the BitsAndBytes library. The following configurations are included in our benchmark:

- **Default (BF16)**: Serving as the baseline.
- **8-bit Quantized**
- **4-bit Quantized**

We also experimented with 2-bit quantization; however, the resulting model failed to generate legible or coherent outputs and was therefore excluded from the evaluation.

### C. Dataset and Query Sampling

We use a diverse set of user queries sampled from the **LMSYS Chat dataset** [16], which reflects real-world usage patterns across a range of topics and prompt styles. The dataset includes prompts involving reasoning, summarization, creativity, factual recall, and instruction-following, enabling a holistic quality assessment.

To experiment with different query sizes, we tokenize the conversations and categorize it into one of the following buckets based on token count:

- **XS**: 0–128 tokens
- **S**: 129–256 tokens
- **M**: 257–512 tokens
- **L**: 513–1024 tokens
- **XL**: 1025–2048 tokens
- **XXL**: 2049–4096 tokens

Each prompt is stored with its corresponding conversation_id to allow consistent tracking and evaluation across stages of the pipeline. For each matchup, a single prompt is shown to two model variants, generating two outputs for evaluation.

### D. LLM-as-a-Judge Evaluation

To determine which model performed better on each prompt, we adopt an **LLM-as-a-judge** approach [17], leveraging strong open-weight LLMs for side-by-side evaluation. We use both LLaMA3.3-70B and LLaMA4-Maverick as judge models.

For each pairwise comparison, the judge is provided with:

- The original user query.
- The two anonymized model outputs (order randomized to mitigate positional bias).
- A system prompt instructing the judge.

*Output Validity Filter:* Before a response is submitted for Elo evaluation, we apply a lightweight validity check using the judge model to exclude corrupted or malformed outputs. The following prompts are used:

Listing 1. System prompt for validity filtering

```
You are a model output validator. Your task is to
determine if the model output is
obviously corrupted. Respond with only "yes" if the
output is corrupted, and only "no"
if it is not. Do not include any other text or
explanation. Corruption will be extremely
obvious, so mostly respond with "no". Look for
issues like incorrect spacing, punctuation
errors, invalid characters, and broken formatting.
```

Listing 2. Prompt template for validity filtering

```
A potentially broken model produced this output:
"""
{output}
"""
Is the output obviously corrupted? Be conservative.
Just respond with no or yes.
```

*Elo Matchup Prompting:* We experimented with dozens of prompt formulations for both the system prompt and the prompt template used to instruct the judge model. While the judge occasionally produces responses that do not strictly adhere to the expected format, the vast majority of outputs are parsable and usable. On average, we observe between 2 and 7 unparsable responses per 1,000 matchups. We also evaluated different input formatting strategies for the judge model, such as providing only the final user message versus the entire conversation serialized as JSON. Although truncated context can be advantageous for short-context models, we observed that including the full input context leads to consistently more accurate and robust judgments, especially in edge cases where prompt ambiguity depends on prior turns.

Listing 3. Elo judge system prompt

```
You are an LLM as judge, comparing two responses
from different LLMs and finding the best answer.
```

Listing 4. Elo judge prompt template

```
Here is the input:
"""
{inp}
"""
Here is response A:
"""
{A}
"""
and here is response B:
"""
{B}
"""
Come to a conclusion on which is the best answer:
"A" or "B". If they are truly equal, respond with
"A=B". Just respond with A or B or A=B. No
justification, nothing, just A or B or A=B. First
letters of your response should be A or B or A=B.
```

*Judge Output Length and Performance Tuning:* We also explored how limiting the judge model's output token budget affects evaluation quality and efficiency. For the validation task—where the judge simply determines if a model output is obviously corrupted—we found that allowing just 3 output tokens is sufficient. The judge typically responds with either `"no"` or a short justification. Since our parser only requires a valid `"no"` to proceed, we can safely truncate longer responses without loss of signal.

In contrast, for Elo matchups, the judge occasionally announces its verdict at the end of a longer response. Truncating these prematurely can lead to loss of usable judgments. Through empirical testing, we found that setting the output token limit to 70 strikes a strong balance between reliability and speed, with only 2–7 unparseable responses per 1,000 prompts—a tolerable rate for large-scale evaluation.

We also experimented with different batch sizes and found that values between 8 and 16 offer optimal throughput for long-context prompts on modern GPU hardware. Constraining the judge's output length from 512 to 3-70 tokens and batch size from 4 to 20, proved to reduce computational overhead, resulting in a 2–4× improvement in evaluation throughput on a 1,000-prompt benchmark.

## E. Elo Scoring and Aggregation

We estimate Elo score differences between model variants based on aggregate win rates obtained from pairwise judgments. Let $w$, $r$, and $t$ denote the number of wins, losses, and ties for the test model, respectively. We compute the expected score as:

$$p = \frac{w + 0.5t}{w + r + t}$$

The Elo difference $\Delta E$ is then calculated as:

$$\Delta E = -400 \cdot \log_{10}\left(\frac{1}{p} - 1\right)$$

This formulation reflects the inverse of the logistic function used in standard Elo rating systems and yields a stable, order-invariant estimate of model quality difference.

## F. Implementation and Computational Budget

All experiments were conducted on `Nvidia H100 GPUs`, with a total compute budget exceeding `2,000 GPU hours`.

Inference was run using `Transformers` [13] with no additional optimizations applied.

To comply with the licensing terms of the LMSYS-Chat-1M dataset, we retained evaluation data—including prompts and model outputs—only for the duration of the research phase. All stored data was permanently deleted after paper submission, in accordance with the dataset's usage agreement. No part of the dataset or its derivatives were shared, redistributed, or used for any purpose beyond this study.

To support reproducibility and enable future research, we have made selected diagrams, configuration files, and non-data-dependent portions of our codebase publicly available at our GitHub repository.[1] No dataset samples or model outputs are included, in accordance with the LMSYS-Chat-1M license agreement.

## IV. RESULTS

We now present our findings on the comparative performance of quantized model variants across different token lengths and prompt complexities. Our analysis is based on Elo deltas computed from over 45,000 pairwise judgments across three quantization configurations—16-bit (BF16), 8-bit, and 4-bit—evaluated using both LLaMA3.3 and LLaMA4 judge models.

## A. Validity of Judge Responses

Before presenting quality comparisons, we evaluate the robustness of the evaluation pipeline itself by measuring the proportion of valid judgments produced by the judge LLMs. As shown in Figure 1, we observe that both `LLaMA3.3` and `LLaMA4` maintain nearly 100% validity for small to medium bucket sizes (XS through M). However, at extreme prompt lengths (XL and XXL), validity drops sharply highlighting

---

[1]https://github.com/samanamp/elo-paper

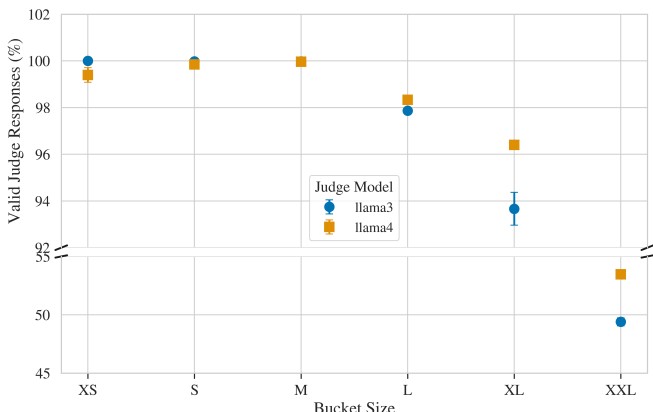

Fig. 1. Percentage of valid LLM-as-a-judge responses across bucket sizes. Validity drops significantly at extreme context lengths (XL, XXL).

the difficulty of generating coherent side-by-side preferences at large context sizes. Based on the fact that `LLaMA4` judge performs better comparatively, we will present rest of the data with `LLaMA4` as judge and in M size data bucket.

### B. Elo Delta by Quantization Level

To analyze the impact of quantization, we focus on matchups between 16-bit, 8-bit, and 4-bit variants. Figure 2 shows the mean Elo deltas across generation lengths for each pairwise matchup under the `LLaMA4` judge, filtered to M-bucket prompts. We observe three key trends:

- The 8-bit variant outperforms the 4-bit model consistently, with growing advantage at longer generation lengths.
- The BF16 (default) model shows increasing separation from 4-bit across token lengths, indicating quality degradation from aggressive quantization.
- Surprisingly, the BF16 vs 8-bit matchups are more balanced, with modest and sometimes negative Elo deltas.

These results suggest that 8-bit quantization preserves quality well in the M-length regime, while 4-bit models suffer significantly—particularly when generating long completions.

### C. Judge Model Sensitivity

To assess the consistency and reasoning capabilities of judge models, we conducted the same evaluation using both `Llama3.3` and `Llama4`. While the `Llama3.3`-based judge appears more sensitive to variations introduced by model quantization, `Llama4-Maverick` exhibits stronger internal coherence in its judgments. Specifically, we expect if Elo(default vs 8bit) $> 0$, then Elo(default vs 4bit) $>$ Elo(8bit vs 4bit).

This transitive relationship holds more reliably under the `Llama4` judge, indicating a higher degree of logical consistency in its evaluations.

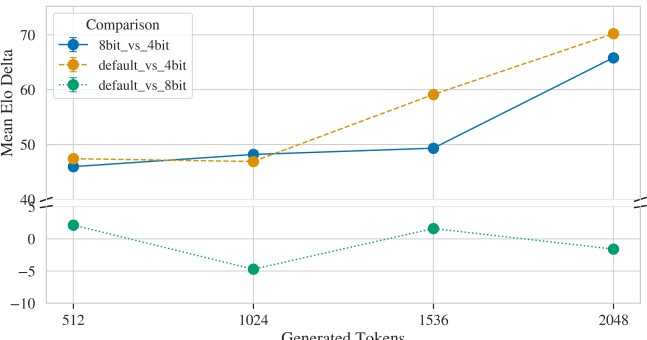

Fig. 2. Elo deltas by token length using `LLaMA4` as judge.

### D. Transitivity and Stability of Elo Deltas

To further assess the logical consistency and convergence behavior of the `LLaMA4-Maverick` judge, we visualize the Elo deltas across increasing numbers of total responses for three key pairwise comparisons: `default vs 4-bit`, `default vs 8-bit`, and `8-bit vs 4-bit`. The figure employs a broken y-axis to simultaneously highlight both large and small-scale variations in Elo differences.

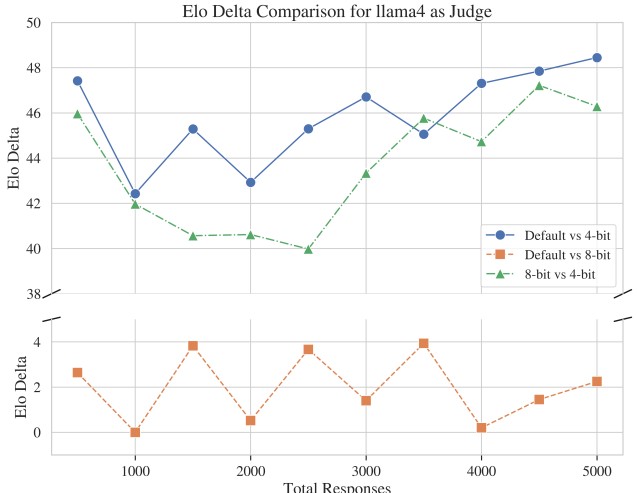

Fig. 3. Elo Delta Comparison with `LLaMA4-Maverick` as the judge model. The broken axis emphasizes the small but consistent deltas between `default` and `8-bit`, while capturing broader trends in other comparisons.

Three key insights emerge from this analysis:

1) **Approximate Transitivity Holds:** The relationship

$$\text{Elo(default vs 4bit)} \approx \text{Elo(default vs 8bit)} + \text{Elo(8bit vs 4bit)}$$

appears to hold consistently across all response sizes, supporting the internal logical coherence of the `LLaMA4` judge's preferences.

2) **No Clear Convergence with More Responses:** Increasing the number of responses beyond the initial

500 does not significantly stabilize the Elo deltas. The lack of convergence suggests that the signal is already reasonably captured at lower response volumes. This reduces the need for costly over-sampling across target, reference, or judge models—saving both inference time and compute resources.

Together, these findings indicate that `LLaMA4-Maverick` offers both logical fidelity and practical efficiency in comparative evaluations.

## V. CONCLUSION AND FUTURE WORK

This paper introduced a scalable, interpretable, and reproducible evaluation framework for benchmarking the effects of efficiency techniques—particularly quantization—on foundation models. By leveraging LLM-as-a-judge comparisons over real-world prompts and computing Elo deltas, we provided a model-agnostic method to assess quality degradation in quantized variants of `google/gemma-3-1b-it`.

Our results highlight three key findings: (1) 256-512 token prompts are best bucket size to compare models; (2) `LLaMA4-Maverick` demonstrates strong internal logical consistency as a judge model, validating transitivity in model preferences; and (3) conducting evaluations on as few as 512 prompts per comparison is sufficient for extracting stable quality signals, enabling meaningful reductions in computational cost.

These insights have practical implications for both model developers and evaluators. Our method reduces reliance on large-scale task-specific benchmarks and allows rapid quality assessments at early development stages using general-purpose judge models and public data.

In future work, we plan to extend this framework in several directions:

- Incorporating pruning and distillation techniques to evaluate trade-offs across a broader range of efficiency optimizations.
- Exploring judge model ensembles to mitigate judgment variance and reduce alignment bias.
- Extending beyond language tasks to include vision-language and code generation benchmarks in multimodal settings.

We hope this work provides a foundation for more rigorous, low-cost evaluations of foundation model efficiency techniques and contributes toward standardizing performance parity benchmarks across the community.

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
