# OpenReview forum: "Benchmarking Efficiency Techniques in GenAI Foundation Models Using an Elo-Based Performance Evaluation Framework"
_iscaconf.org/ISCA/2025/Workshop/MLArchSys — MLArchSys 2025 Poster_

### Official Review · Reviewer_rNUJ · 2025-05-16
**Clear rejection due to lack of novelty and validation**

**Confidence:** 4
**Rating:** 3

**Detailed Feedback And Questions For Authors:**

The paper proposes an Elo-based evaluation framework using LLM-as-a-judge to assess the impact of quantization on foundation models. While the methodology is clearly described and leverages recent trends in preference-based evaluation, the contribution is limited in scope and originality. The experiments focus solely on a single model (Gemma-3B) and one efficiency technique (quantization), despite the broader claim of benchmarking “efficiency techniques.” There is no validation against human judgments, no statistical analysis of judgment reliability, and no comparison to existing evaluation baselines. As such, the work does not meet the novelty or rigor expected for publication. To improve, the authors should expand the experimental scope, include human-grounded validation, and clarify how their approach advances beyond existing efforts like LMSYS Arena or MT-Bench.

**Top Reasons To Accept The Paper:**

The paper addresses an important and underexplored challenge: standardized evaluation of efficiency techniques (e.g., quantization) in foundation models.

The use of real-world prompts and systematic judge validation contributes toward practical benchmarking.

**Top Reasons To Reject The Paper:**

The core contribution—applying Elo scoring via LLM-as-a-judge for evaluating quantized models—is incremental and lacks novelty. Similar approaches (e.g., Chatbot Arena, MT-Bench) already exist and are better validated.

The scope is too narrow: the experiments are limited to only one model (Gemma-3B) and quantization (no pruning or distillation), undermining the “benchmarking efficiency techniques” claim in the title.

There is no rigorous analysis of evaluation reliability (e.g., judge disagreement rates, inter-rater consistency, calibration studies), which is critical when using LLMs for subjective judgments.

---

### Official Review · Reviewer_z3pp · 2025-05-18
**Review summary**

**Confidence:** 5
**Rating:** 4

**Detailed Feedback And Questions For Authors:**

This paper proposes an Elo-based scoring framework for LLM evaluation to address the challenge of assessing model quality parity after efficiency optimizations that alter model’s numeric.

- Pros

1. Experiments using Gemma are conducted to demonstrate the framework functionality.
2. It is valuable that the authors show the difference between judge models and their transitivity and stability of Elo scores.

- Cons

1. One of the main motivations of this work is to enhance the reliability of existing evaluation methods. However, no clear data points are presented to support this. Including results that demonstrate the inadequacy (e.g., lack of sensitivity) of existing evaluations for the same quantized models would be helpful.
2. Insights are specifically for one test model and it is unknown how general they would be.
3. Quantization details are unclear for the 8/4-bit quantized version. Is it for weight or activation?

**Top Reasons To Accept The Paper:**

None

**Top Reasons To Reject The Paper:**

None

---

### Official Review · Reviewer_3QDL · 2025-05-18
**Review for paper**

**Confidence:** 4
**Rating:** 6

**Detailed Feedback And Questions For Authors:**

- In the results section, transitivity is used in two different ways. The first is x>y and y>z imply x>z, and the other is $x \approx y$ and $y \approx z$ imply $x \approx z$. I think the approximate transitivity needs a little more care, because the two errors on the relationships $x \approx y$ and $y \approx z$ will accumulate to some degree. I would suggest to quantify this a little more.
- "No Clear Convergence with More Responses" -- it would be interesting to see more data here, such as the standard deviation / error bars at each point. This is actually a bit concerning because it means the evaluation may not be very stable, so it would be good to have a more detailed understanding of why this is happening. It could mean something needs to be adjusted in the way the Elo scores are calculated, or just more data is required for stabilization. If we want to be able to detect smaller differences convergence would be a desirable property. However, the types of differences measured here (4 bit vs 8 bit vs bf16) are likely more coarse so for this experiment it's probably good enough just to show the differences. I would suggest to change the wording to something like: For the experiments here, 500 samples was sufficient to detect statistically meaningful differences (try to actually verify this), etc.

**Top Reasons To Accept The Paper:**

The paper sets up a framework for an automated way to compare different quantization schemes for LLMs using a judge LLM to create Elo scores. This may help evaluating quantization degradation in cases where other benchmarks are less reliable.

**Top Reasons To Reject The Paper:**

The main question for publication in my view is what novelty the paper introduces beyond what is already available through existing blind comparisons frameworks. I would like to better understand what novelty was required especially for using this in the context of efficiency techniques. It would also be useful to show cases where specific benchmarks / metrics fail to capture something that the Elo score is able to.